# Hydriding, Oxidation, and Ductility Evaluation of Cr-Coated Zircaloy-4 Tubing

**Yong Yan \*, Tim Graening and Andrew T. Nelson**

Oak Ridge National Laboratory, Oak Ridge, TN 37831, USA
* Correspondence: yany@ornl.gov

**Abstract:** Accident-tolerant fuel concepts have been developed recently in diverse research programs. Recent research has shown clear advantages of Cr-coated Zr cladding over bare cladding tubes regarding oxidation behavior under the design basis loss-of-coolant accident condition. However, limited data are available about the hydriding behavior of the Cr coating. For that purpose, Cr-coated Zricaloy-4 tubes were tested to investigate the effects of hydriding, oxidation, and postquench ductility behavior on coated Zr cladding. A high-power impulse magnetron sputtering (HiPIMS) process was used to produce a high-density coating on the Zircaloy-4 tube surface. Coated and uncoated Zircaloy-4 tube specimens underwent one-sided hydriding in a tube furnace filled with pure hydrogen gas at 425 °C. The tubing specimen ends were sealed with Swagelok plugs before the hydriding runs. For uncoated specimens, H analysis of the hydrided specimens indicated that the H content increased as the test time and initial pressure increased. However, almost no change was observed for the coated specimens that were hydrided under the same test conditions. After one-sided hydriding, the hydrided coated and uncoated specimens were exposed to steam at high temperatures for two-sided oxidation studies to simulate accident conditions. The coated specimens showed a slower oxidation: oxygen pickup was 50% lower than the uncoated specimens tested under the same conditions. Ring compression testing was performed to evaluate the embrittlement behavior of the Cr-coated specimens after hydriding and oxidation. The results indicated that the HiPIMS coating provides excellent protection from hydriding and oxidation at high temperatures.

**Keywords:** accident-tolerant fuel; LOCA; hydriding; oxidation; Cr-coating; Zircaloy-4; ductility; mechanical property

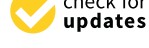



## 1. Introduction

The Fukushima accident in March of 2011 motivated research on accident-tolerant nuclear fuels (ATF) [1,2]. Efforts to develop a new ATF cladding concept emphasized the goal of enhancing the cladding's high-temperature performance and increasing the response time in case of a severe accident scenario. Potential enhancements were considered, including a thin coating of highly corrosion-resistant material applied to the cladding's surface. Coated zirconium cladding concepts have become prioritized approaches of industry ATF efforts [3]. Licensure of coated zirconium cladding should be less costly and time-consuming than FeCrAl or SiC/SiC; this strategy is logically consistent because coating failure would result in the same base cladding concept that is already licensed (assuming that no credit is taken for coating performance) [4].

The purpose of this thin coating is to improve corrosion, hydriding, and wear performance without interfering with the neutronic or mechanical performance of the base cladding under normal operating conditions. Different coating materials, thicknesses, coating processes, process parameters, and testing methods affect the cladding's microstructure and mechanical properties. Initial deployment of coated Zr cladding is also envisioned to be simpler than FeCrAl or SiC/SiC concepts if it is assumed that the worst-case scenario would be failure or delamination of the coating. This scenario would leave the cladding

material as the underlying Zr alloy, which is presumably already licensed [5]. Even though many methods have been developed to investigate coatings for cladding materials and their mechanical properties [6,7], including the oxidation resistance of Cr coatings [8–13], a general understanding of hydriding behavior of Cr-coated cladding is missing. In this work, Cr-coated claddings and bare cladding were compared by hydriding them to different target hydrogen levels. Then the claddings were steam oxidized and mechanically tested.

## 2. Materials and Methods

All the experiments were conducted with pressurized water reactor (PWR) Zircaloy-4 tubing, fabricated by Cameco Fuel Manufacturing (Port Hope, ON, Canada). The tubing is consistent with $17 \times 17$ fuel assembly specifications. It has an outer diameter of 9.50 mm and a wall thickness of 0.56 mm. The material was stress-relief annealed and conformed to the typical geometry of a PWR cladding. The compositions of the Zircolay-4 tubes used in this work were measured and chemical composition ranges are provided in Table 1.

**Table 1.** Measured chemical composition of the ORNL Zircaloy-4 cladding tubes.

| Parameter | ORNL Zircaloy-4 |
| --- | --- |
| Zr (wt %) | Balance |
| Sn (wt %) | 1.29–1.37 |
| Fe (wt %) | 0.30–0.34 |
| Cr (wt %) | 0.10 |
| Ni (wt %) | - |
| H (wppm) | 4–6 |

The tube's surface roughness was measured using a Mahr (Göttingen, Germany) MarTalk profilometer with the MarWin software package version 5.0. The tubes were measured in different areas to identify possible differences. Roughness measurements were performed at 11,200 points across a 4 mm length profile on the tube surface. Even though the average surface roughness was within the specified limits of 0.81 μm, maximum roughness peaks of approximately 2 to 4 μm or more were observed near scratches on the surface. These scratches were concerning because they could affect coating adherence or cause cracks or discontinuities in the coating. Therefore, the as-received tubes were subsequently polished using 30 μm polishing paper down to 3 μm polishing paper before coating, hydriding, and steam oxidations. Polishing improved the surface roughness from an average roughness of around 0.36 to around 0.1 μm and decreased maximum roughness from 4 to around 1 μm. Surfaces of as-received and polished Zircaloy-4 tubes are shown in Figure 1.

After polishing, Cr coatings were applied on the Zircaloy-4 surface using a high-power impulse magnetron sputtering (HiPIMS) process by Acree Technologies, Inc. (Concord, CA, USA), which enables higher densities of coatings than conventional physical vapor deposition methods. The polished Zircaloy-4 tubes were first cleaned using an ultrasonic cleaner and isopropanol, followed by plasma cleaning. The power density was 9.9 W/cm$^2$. During the first 10 min of the coating process, a voltage bias of $-1000$ V was applied to the substrate to increase bonding strength. Afterward, the bias voltage during the coating process was set to $-100$ V. The temperature was maintained at approximately 200 °C throughout the entire process. The objective of coating at elevated temperature was to introduce compressive stresses, which help to offset the tensile stress during operation under PWR conditions and potential accident scenarios. The HiPIMS process was utilized to create a dense and well adherent coating with a thickness between 6 and 7 μm. Cross sectional SEM images with different magnifications are shown in Figure 2 to provide a general overview and to highlight the grain structure of the Cr coating. A thin coating was chosen to mitigate its effect on the mechanical properties compared with uncoated material

while allowing investigation of the oxidation behavior and hydrogen pickup. Because of the thin coating, the mechanical behavior after oxidation is rooted in the reduced oxidation and potentially reduced hydrogen pickup instead of an increased circumferential thickness.

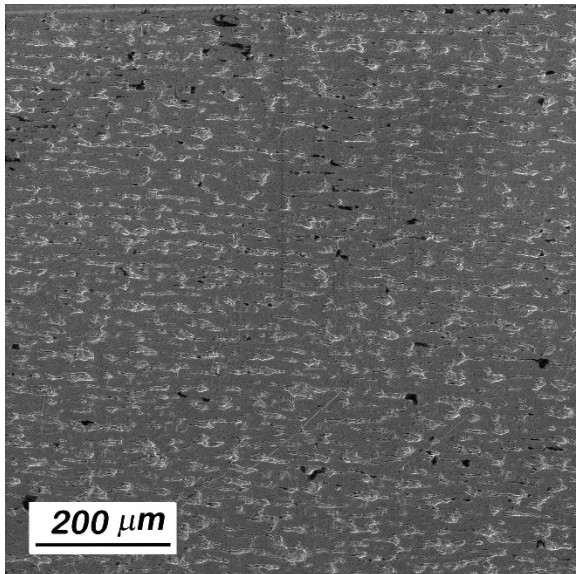
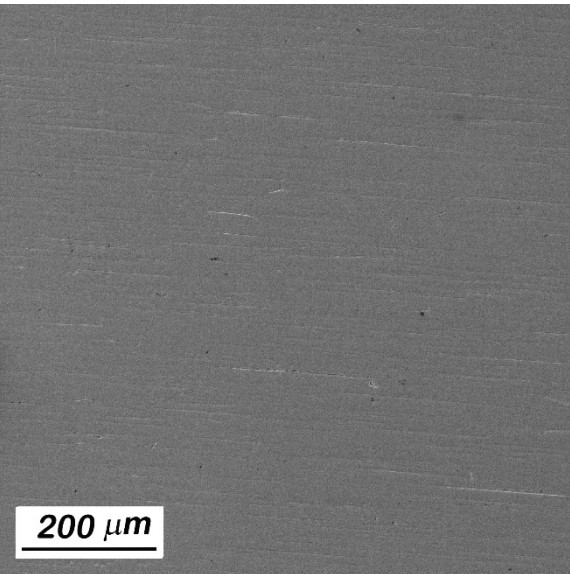

**Figure 1.** SEM images of the surfaces of as-received Zircaloy-4 tubes (**right**) and polished Zircaloy-4 tube (**left**).

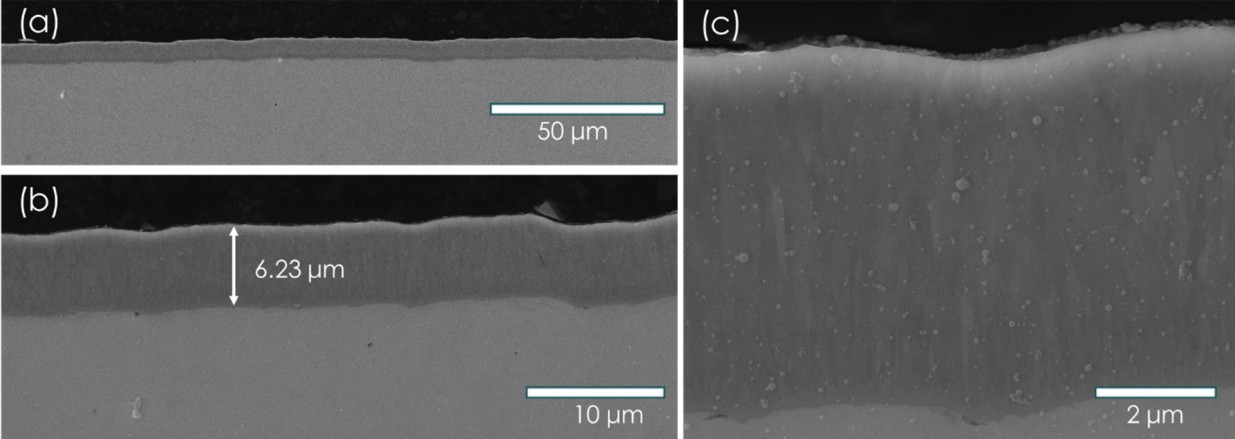

**Figure 2.** Cross sectional SEM images in secondary electron mode with SEM images: (**a**) low magnification image, (**b**) thickness measurement image, and (**c**) high magnification image showing the directional grain growth and good adherence of the coating.

The hydriding system consists of a tube furnace with a stainless steel tube that is heated to high temperatures in the presence of hydrogen to introduce a desired quantity of hydrogen into the specimen [14]. Hydrogen uptake at elevated temperature results in the formation of hydrides, which are stable when cooled to room temperature. The tube furnace has a 48 cm long heating zone. Hydrogen is supplied by an Avantor hydrogen generator (Avantor Inc., Radnor, PA, USA). The stainless steel tube was evacuated to $5 \times 10^{-2}$ Torr and then purged twice using Ar gas before the hydrogen was introduced. The hydriding temperature was set to 425 °C, and the initial tube pressure was varied to adjust to the specimen's target hydrogen concentrations. The hydrogen content of the hydrided specimens was measured using the inert gas fusion technique per ASTM E1447-05

or the vacuum hot extraction technique per ASTM E 146-83. The error of hydrogen contents reported in this study is about 10 wppm.

Steam oxidation tests were performed using a resistance-heating furnace in flowing steam in the ambient atmosphere. The steam oxidation system consists of a 122 cm long tube furnace with a uniform temperature zone that is longer than 10 cm. A quartz holder held the sample. The desired heating and cooling rates were achieved via controlled movements of the specimens into and out of the furnace. In this work, the heating rate is about 20 °C/s to within 200 °C of the target temperature (1000 and 1205 °C) and a few degrees per second thereafter. The average cooling rate from the hold temperature to the 800 °C quench temperature is about 10 °C/s. The samples were oxidized via the following process. (1) The samples were rapidly heated to the transient temperature of 1000 and 1205 °C. (2) The samples were held at the transient temperature for various hold times. (3) The samples were cooled to 800 °C. (4) The samples were water quenched. The test parameters, such as heating rate, cooling rate, hold temperature, and water quench temperature were carefully selected based on a guideline provided by the US Nuclear Regulatory Commission [15]. The test time was determined by oxygen pickup near the ductile-to-brittle transition point of the oxidized specimens. Figure 3 shows a typical temperature history for the steam oxidation tests with Zircaloy-4 at 1000 °C. The oxidation testing procedure was detailed by Yan et al. [16,17]. To determine sample weight gain, sample mass was directly measured before and after oxidation testing using a balance with ±0.1 mg accuracy. Sample mass was then normalized to the steam-exposed surface area of the specimen.

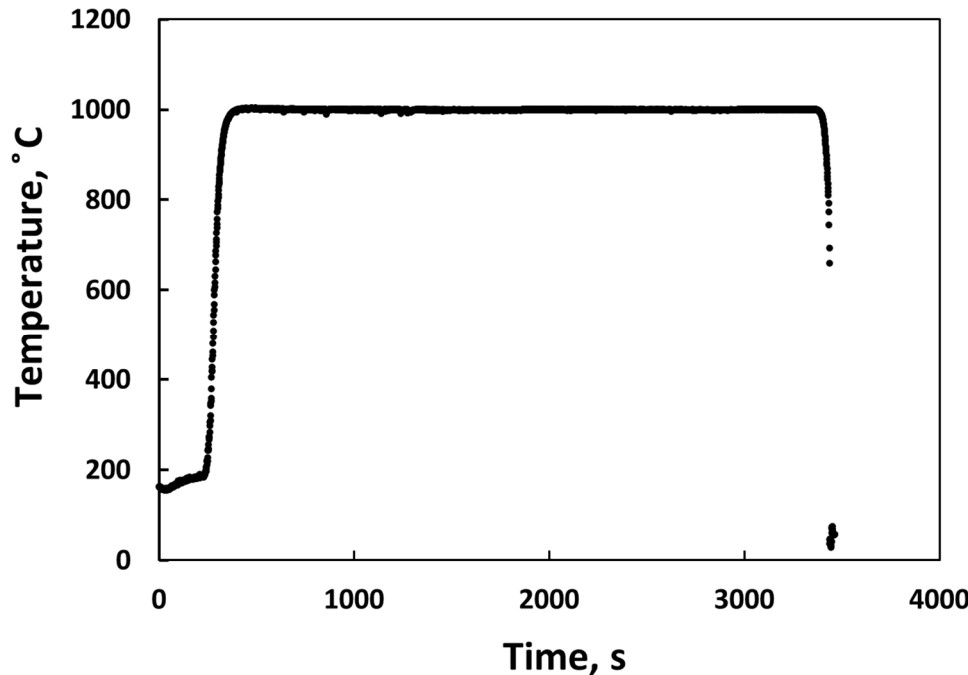

**Figure 3.** Temperature history for the steam oxidation tests with Zircaloy-4 at 1000 °C for 3000 s.

Ring compression tests (RCTs) were used to evaluate the remaining ductility of hydrided and postoxidation samples. An Instron 5966 material test system was used to perform the RCT. The instrument's calibration has been verified annually for measurement of loads by the load cell, the determination of crosshead displacement, and the determination of crosshead speed. The crosshead displacement rate for RCTs was set to 0.033 mm/s (2 mm/min). The test was conducted in a displacement-controlled mode, as described previously [18]. To end the test, the compressive load was released as soon as a sharp load decrease occurred, as shown in Figure 4.

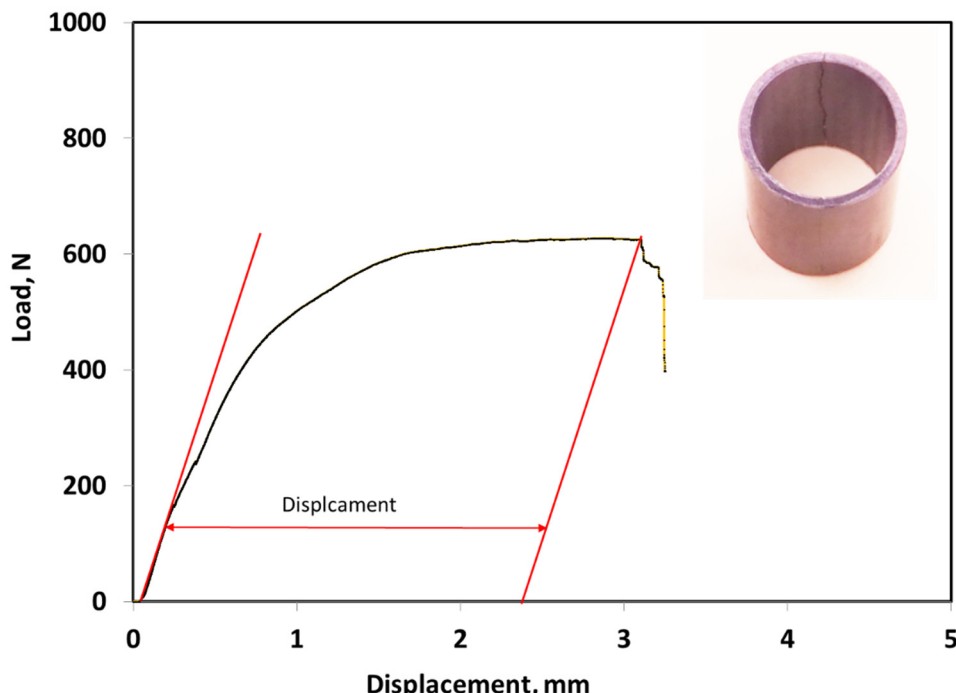

**Figure 4.** Load–displacement curve for Cr-coated Zircaloy-4 oxidation sample tested at 2 mm/min compression rate. The oxidation test was conducted at 1000 °C for 3000 s. The insert is a posttest RCT sample showing a through-wall crack that corresponds to a sharp load decrease.

To examine whether the Cr coating protects against hydrogen charge, one-sided hydriding was performed using both uncoated and coated tubes. The hydriding sample length was 76.2 mm. Each sample was cleaned with water and then placed in isopropanol in an ultrasonic bath for about 200 s. After the sample was completely dried, three 25.4 mm long zirconia pellets were inserted into the sample to reduce the gas volume inside the sample. Both ends of the sample were then sealed with Swagelok end plugs. The sealed sample was placed into the stainless steel tube center for hydriding.

The target hydrogen concentrations ranged from 150 to 750 wppm, controlled by the initial hydrogen gas pressure, test temperature, and test time. Coated and uncoated specimens were hydrided under the exact same conditions to assess repeatability. The hydrogen content of the hydrided test specimens was measured via a destructive method to provide the mass of hydrogen per mass of Zircaloy-4 in units of weight parts per million. Following the hydriding experiment, a 1 mm long ring was cut from the center of each hydrided sample for hydrogen analysis. Then 10–12 mm long ring samples were sectioned from areas adjacent to the 1 mm center ring for future testing.

### 3. Results and Discussion

*3.1. Materials Hydriding and Characterization*

Vacuum hot extraction as employed in the present work can only accept crucible dimensions of approximately 1 cm diameter by 1 cm length, so the hydrogen concentration can be measured using a whole 1 mm long ring. When the hydrogen analysis was performed on hydrided samples, the as-fabricated coated and uncoated Zircaloy-4 specimens were also measured as a reference. The sample labeling and results of the hydriding measurements are summarized in Table 2. The hydrogen content of the uncoated specimens gradually increased with increasing initial pressure and hold time. However, the hydrogen content of the coated specimens remained lower than 30 wppm, indicating that the coating provided excellent protection against hydriding. Atmospheric humidity trapped inside the Swagelok-sealed tube specimens might have caused the small hydrogen pickup on the coated specimens.

**Table 2.** Hydriding and hydrogen measurements of coated and uncoated Zircaloy-4.

| Sample | Materials | Hydriding Temp. (°C) | Pressure (Torr) | Time (h) | Measured H (wppm) | Comments |
|---|---|---|---|---|---|---|
| U0 | Uncoated | N/A | N/A | N/A | 10 | As-polished |
| C0 | Coated | N/A | N/A | N/A | 9 | As-coated |
| U1 | Uncoated | 425 | 33 | 50 | 143 | Target: 150 wppm |
| C1 | Coated | 425 | 33 | 50 | 26 | Repeat Test U1 |
| U2 | Uncoated | 425 | 51 | 50 | 319 | Target: 300 wppm |
| C2 | Coated | 425 | 51 | 50 | 24 | Repeat Test U2 |
| U3 | Uncoated | 425 | 71 | 51 | 755 | Target: 750 wppm |
| C3 | Coated | 425 | 71 | 51 | 32 | Repeat Test U3 |

Figure 5 shows hydrogen pressure profiles for the one-sided hydriding Test C3 (coated tubing) and Test U3 (uncoated tubing) Zircaloy-4 samples. Figure 6 shows the temperature profiles corresponding to the two hydriding tests in Figure 5. For the uncoated sample, the gas pressure initially increased with increasing temperature. When the temperature was near the hold temperature of 425 °C, the pressure increase rate slowed, and then pressure started decreasing because hydrogen was absorbed by the Zr specimens at elevated temperatures. The hydriding process stopped when the furnace power was shut down. For the coated sample, the gas pressure increases with increasing temperature. Constant pressure was maintained at the hold temperature during the hydriding process, indicating that the coating provides excellent protection against hydriding under the test condition used here. The pressure decrease at the end of the test was due to a temperature decease. Using different initial hydrogen gas pressures and test durations, various hydrogen contents were achieved for the uncoated Zircaloy-4 samples. These tests were repeated with coated samples.

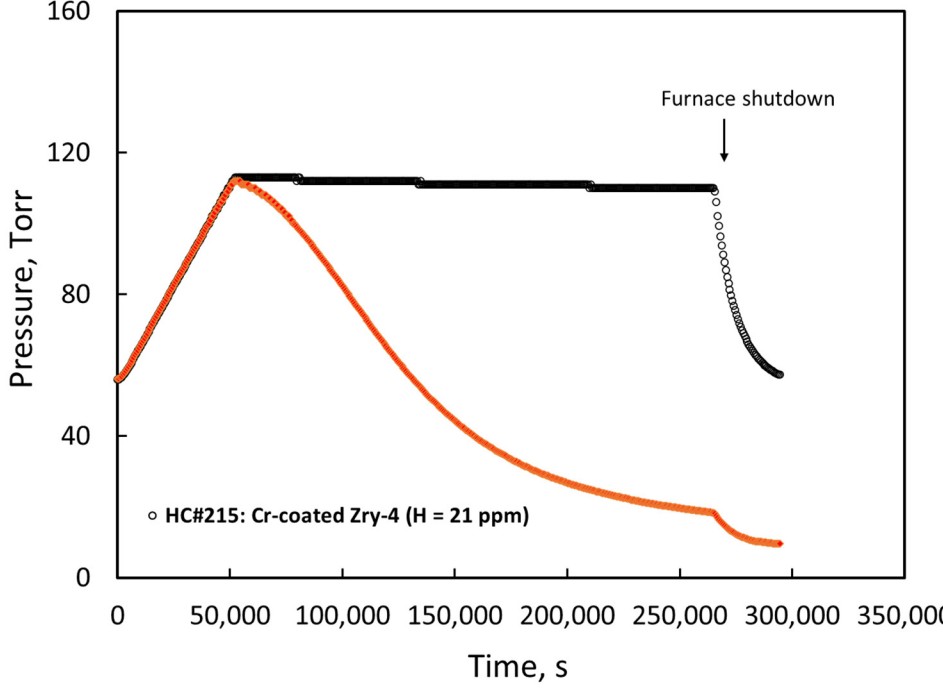

**Figure 5.** Hydrogen pressure profiles for one-sided hydriding of Cr-coated (C3) and as-received (U3) Zircaloy-4 samples.

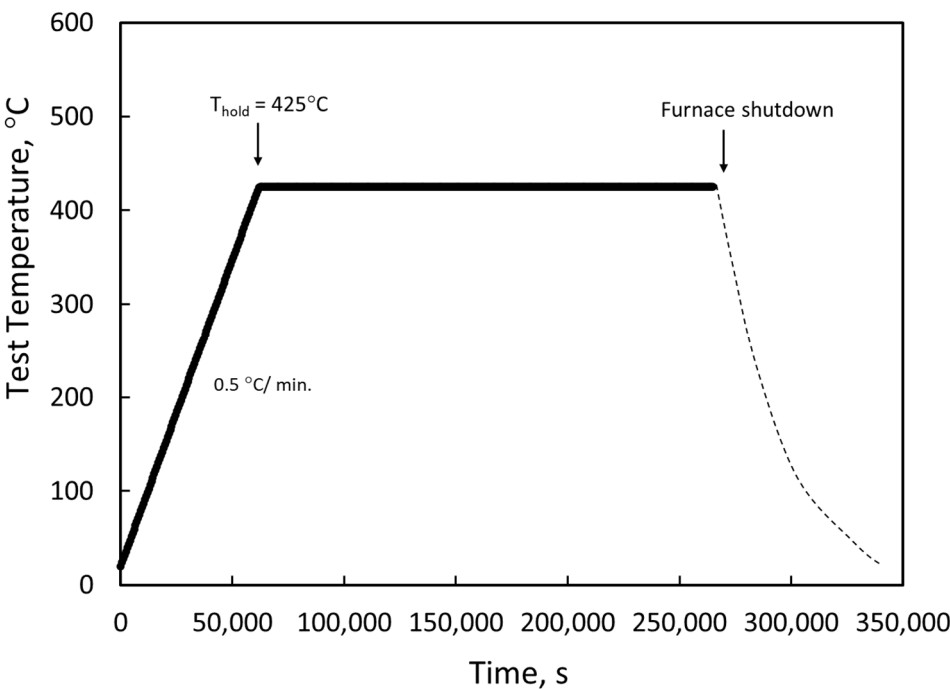

**Figure 6.** Temperature profiles for one-sided hydriding of Cr-coated (C3) and as-received (U3) Zircaloy-4 samples.

Metallographic examinations were performed for the hydride morphology after hydriding tests. Although one-sided hydriding was performed, the hydriding procedure resulted in a uniform distribution of circumferential hydrides for all uncoated specimens. No hydride rim was observed near the outer surface. As shown in Figure 7a–c, for the uncoated samples, the hydride density increases with increasing sample hydrogen concentration. This result is consistent with the measured hydrogen content shown in Table 2. The dark lines in Figure 7 are not microcracks. They are hydrides that appeared after the hydriding experiments. Imaging the hydrides for coated samples was difficult because their hydrogen contents were very low.

### 3.2. High Temperature Steam Oxidation Tests

Two-sided high-temperature steam oxidation was performed with coated and uncoated samples before and after hydriding. Before the steam oxidation test, the oxidation system was evaluated with nuclear grade commercial Zircaloy-4 cladding, and the weight gain (the sample mass difference before and after the steam oxidation tests) and oxide layer thickness test were compared with the Cathcart–Pawel predicted values [19]. For temperature evaluation testing, one thermocouple was directly welded onto the outer surface of the sample. Another thermocouple was exposed to steam at an axial position 1.0 cm away from the welded thermocouple. The sample weight gains of the evaluation tests are in excellent agreement with the calculated weight gains using the Cathcart–Pawel correlation. The error percentage of the measured weight gain relative to Cathcart–Pawel calculated weight gain is less than 5%, indicating that the oxidation system was well calibrated.

For each oxidation test, the coated and uncoated specimens were placed onto a sample holder next to each other. These two samples were hydrided under the same hydriding conditions (see Table 2) and were also oxidized in one oxidation run to ensure a direct comparison between the coated and uncoated specimens. After steam oxidation testing, the sample weight gain was measured. Tables 3 and 4 summarize the weight gain values for the specimens oxidized in steam at 1205 and 1000 °C, respectively. The measured specimen weight gain of the coated specimens was significantly lower than the uncoated tubing specimens—approximately 50% of the uncoated samples' weight gain. Therefore,

the coating provides significant protection against oxidation on the outer surface at 1205 and 1000 °C for test oxidation times performed in this work.

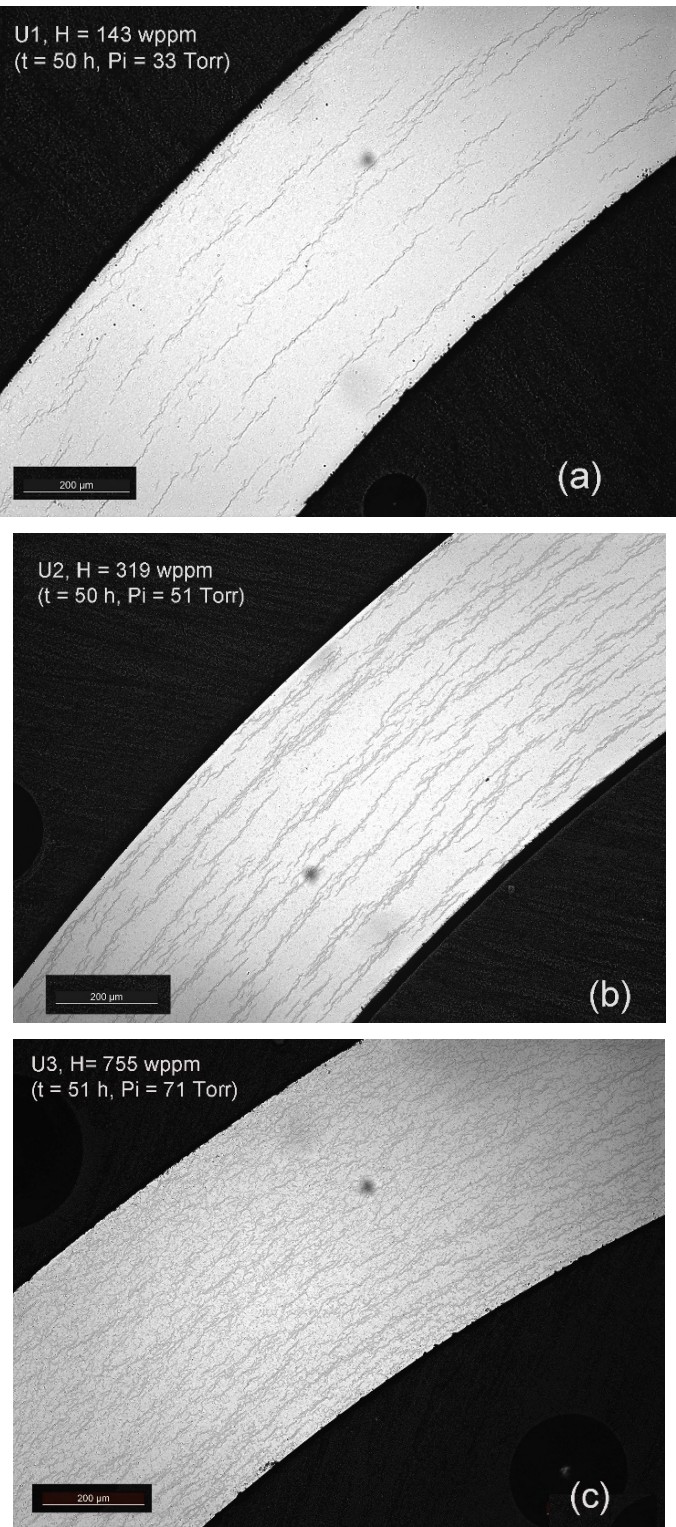

**Figure 7.** Micrographs showing hydride (dark lines) distributions in one-sided hydrided specimens with the uncoated Zricaloy-4 at 425 °C for hydrogen concentrations of (**a**) 143 wppm, (**b**) 319 wppm, and (**c**) 755 wppm. The test conditions for each specimen are given in Table 2. The hydrogen contents given were measured using the vacuum hot extraction method.

**Table 3.** Weight gain values for coated and uncoated Zircaloy-4 oxidized in steam at 1205 °C, cooled to 800 °C, and then water quenched.

| Sample | Measured H * (wppm) | Oxidation Temp. (°C) | Hold Time at Target Temp. (s) | Measured Weight Gain (mg/cm²) | Comments |
|---|---|---|---|---|---|
| ox1205-U0 | 10 | 1205 | 100 | 8.7 | As-polished |
| ox1205-C0 | 9 | 1205 | 100 | 4.5 | As-coated |
| ox1205-U1 | 143 | 1205 | 100 | 8.6 | Hydrided for 50 h, $P_i$ = 32.7 Torr |
| ox1205-C1 | 26 | 1205 | 100 | 4.5 | Hydrided for 50 h, $P_i$ = 32.8 Torr |
| ox1205-U2 | 319 | 1205 | 100 | 8.9 | Hydrided for 50 h, $P_i$ = 51.1 Torr |
| ox1205-C2 | 24 | 1205 | 100 | 4.6 | Hydrided for 50 h, $P_i$ = 51.0 Torr |
| ox1205-U3 | 755 | 1205 | 100 | 8.6 | Hydrided for 51 h, $P_i$ = 71.3 Torr |
| ox1205-C3 | 32 | 1205 | 100 | 4.5 | Hydrided for 51 h, $P_i$ = 71.3 Torr |

* Hydrogen contents were measured before oxidation tests. $P_i$ is initial pressure.

**Table 4.** Weight gain values for coated and uncoated Zircaloy-4 oxidized in steam at 1000 °C, cooled to 800 °C, and then water quenched.

| Sample | Measured H * (wppm) | Oxidation Temp. (°C) | Hold Time at Target Temp. (s) | Measured Weight Gain (mg/cm²) | Comments |
|---|---|---|---|---|---|
| ox1000-U0 | 10 | 1000 | 3000 | 12.0 | As-polished |
| ox1000-C0 | 9 | 1000 | 3000 | 5.9 | As-coated |
| ox1000-U1 | 143 | 1000 | 3000 | 11.8 | Hydrided for 50 h, $P_i$ = 32.7 Torr |
| ox1000-C1 | 26 | 1000 | 100 | 5.9 | Hydrided for 50 h, $P_i$ = 32.8 Torr |
| ox1000-U2 | 319 | 1000 | 3000 | 12.1 | Hydrided for 50 h, $P_i$ = 51.1 Torr |
| ox1000-C2 | 24 | 1000 | 3000 | 6.1 | Hydrided for 50 h, $P_i$ = 51.0 Torr |
| ox1000-U3 | 755 | 1000 | 3000 | 12.3 | Hydrided for 51 h, $P_i$ = 71.3 Torr |
| ox1000-C3 | 32 | 1000 | 3000 | 6.0 | Hydrided for 51 h, $P_i$ = 71.3 Torr |

* Hydrogen contents were measured before oxidation tests. $P_i$ is initial pressure.

　　　Metallographic examinations were performed for selected postoxidation samples. Figure 8 shows micrographs of a pair of coated and uncoated Zircaloy-4 specimens tested together at 1000 °C for 3000 s. No zirconium oxide layer was observed on the coating surface or beneath the coating, confirming that the coating provides significant protection. Figure 9 shows optical images of a pair of coated and uncoated Zircaloy-4 specimens tested together at 1205 °C for 100 s, showing that the coating provides significant protection from steam oxidation at temperatures up to 1205 °C.

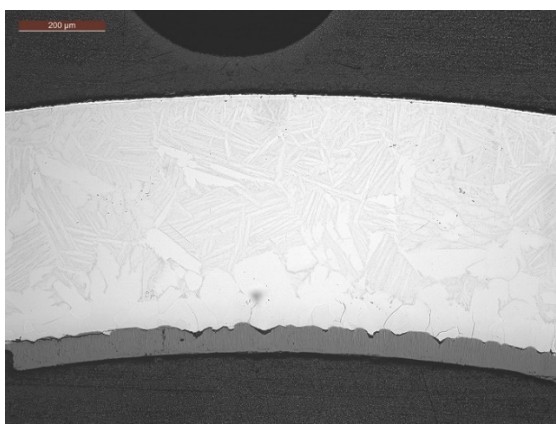
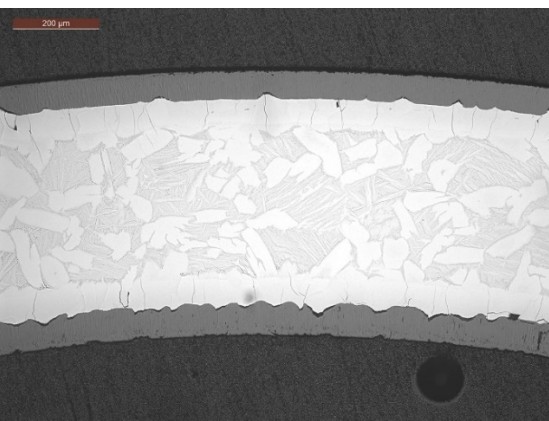

**Figure 8.** Micrographs of (**left**) Cr-coated and (**right**) uncoated Zircaloy-4 oxidized in steam at 1000 °C for 3000 s. No zirconium oxide layer was observed on the coating surface or beneath the coating.

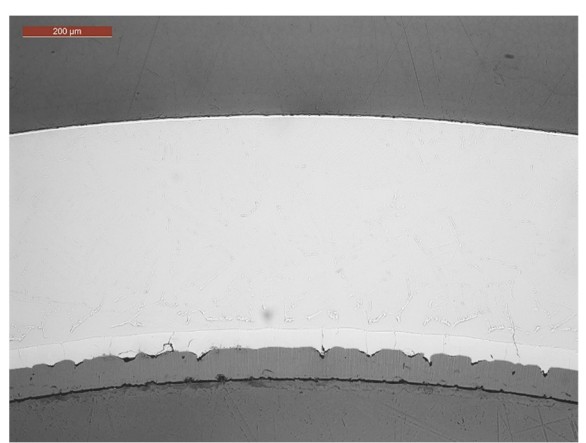
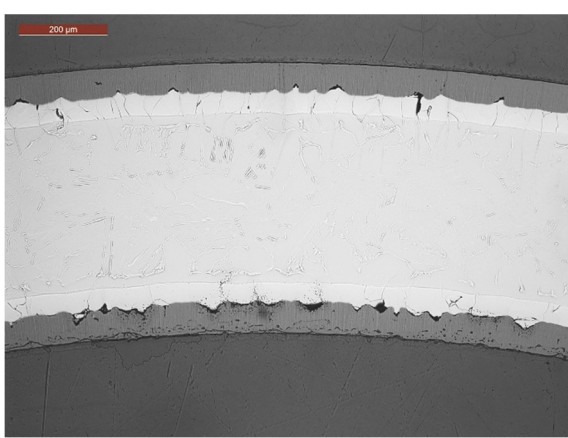

**Figure 9.** Micrographs of (**left** Cr-coated and (**right**) uncoated Zircaloy-4 oxidized in steam at 1205 °C for 100 s. No zirconium oxide layer was observed on the coating surface or beneath the coating.

*3.3. Ring Compression Testing of Hydrided and Oxidized Zircaloy-4 Specimens*

RCTs were performed with 10 mm long rings sectioned from hydrided Zircaloy-4 samples with various hydrogen contents. Lateral compression of the hydrided samples by the RCT yielded a load–displacement curve used to evaluate the ductility of the hydrided samples. The load–displacement curves were analyzed by the offset-displacement method that is the standard RCT measure of ductility [18]. In this work the offset strain was determined from load–displacement data by being normalized to the as-fabricated outer diameter (9.50 mm) to give a nominal plastic hoop strain. For ease of comparison, multiple load–displacement curves are placed on a single graph. The curves are shifted along the *x*-axis but reflect the maximum load and change in displacement. For the uncoated samples, the plots in Figure 10 demonstrate that ductility remains almost the same as the as-fabricated tubing until the hydrogen concentration reaches 320 ppm, but the ductility is significantly reduced when hydrogen pickup is 750 ppm. However, ductility does not change for the coated specimens that were hydrided under the same conditions as the uncoated specimen (Figure 11), which confirms that the coating provides protection from hydriding at 450 °C in pure hydrogen gas.

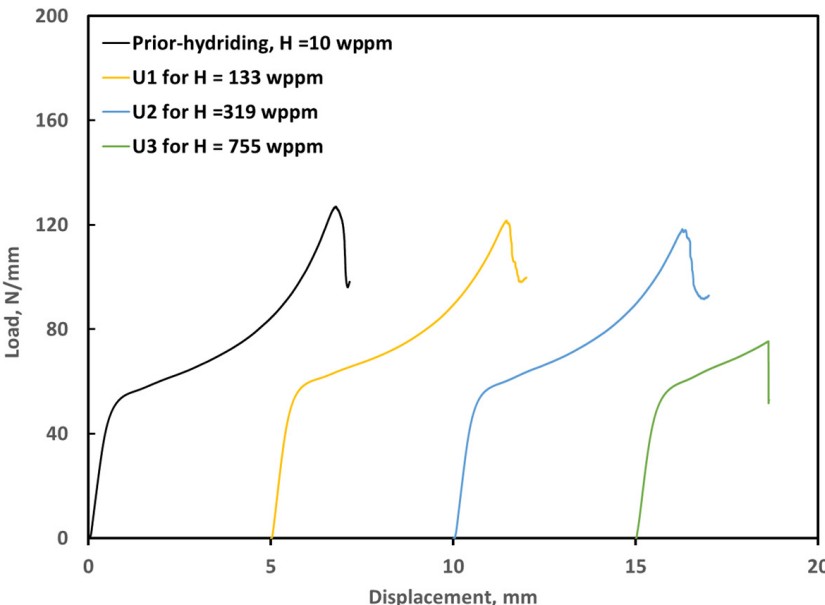

**Figure 10.** Ring-compression test load–displacement data of uncoated Zircaloy-4 hydrided at 425 °C for ~50 h. The initial pressure varies from 33 to 71 Torr.

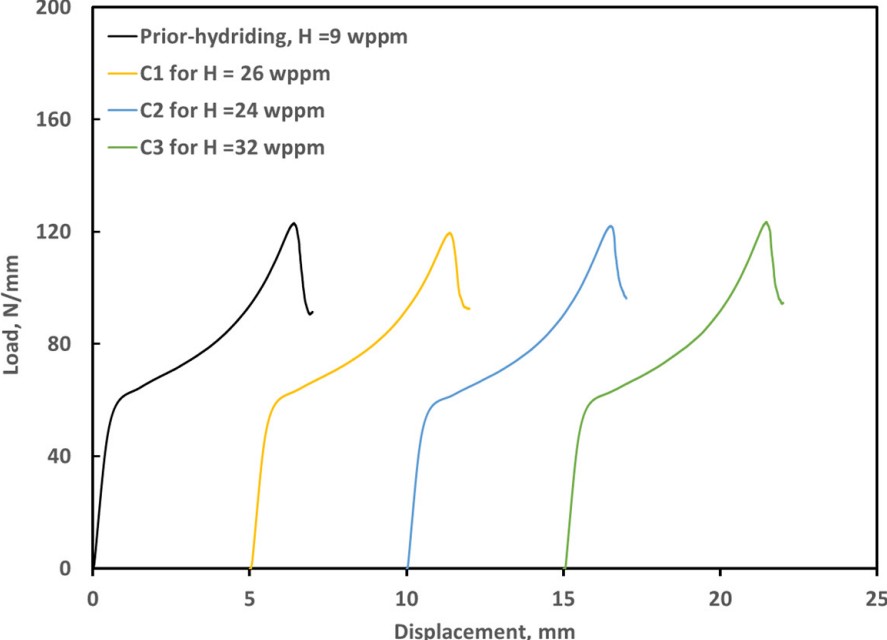

**Figure 11.** Ring-compression test load–displacement data of coated Zircaloy-4 hydrided at 425 °C for ~50 h. The initial pressure varies from 33 to 71 Torr.

Following the steam oxidation tests of the hydrided coated and uncoated Zircaloy-4 specimens, RCTs were performed to evaluate the ductility of the oxidized specimens [7]. For 1000 °C oxidation specimens, the RCT was performed at 20 °C. For 1205 °C oxidation specimens, the RCT was performed at 135 °C. Testing stopped when a considerable load decrease was observed, which was taken as an indication of cracking. Table 5 summarizes the postquench ductility results for prehydrided coated and uncoated Zircaloy-4 specimens oxidized at 1205 °C for 100 s. To better demonstrate the offset strain trend, the RCT results are plotted for the uncoated and coated specimens in Figures 12 and 13. For uncoated materials, the displacement and maximum load of the oxidized specimens

reduced dramatically with increasing hydrogen content (Figure 12). For coated materials, the RCT results changed only slightly after hydriding and oxidation tests, as shown in Figure 13. The shapes of the plots almost remained the same, and no ductility reduction was observed for any of the oxidized coated samples examined in this work. This result confirms that the coating provides excellent protection from both hydriding and oxidation. Table 6 summarizes the postquench ductility results for pre-hydrided coated and uncoated Zircaloy-4 oxidized at 1000 °C for 3000 s.

**Table 5.** RCT results for coated and uncoated Zircaloy-4 oxidized in steam at 1205 °C for 100 s.

| Sample | Measured H * (wppm) | Weight Gain (mg/cm$^2$) | Offset Displace-ment(mm) | Offset Strain (%) | Comments |
|---|---|---|---|---|---|
| ox1205-U0 | 10 | 8.7 | 0.92 | 9.7 | As-polished |
| ox1205-C0 | 9 | 4.5 | 1.88 | 19.7 | As-coated |
| ox1205-U1 | 143 | 8.6 | 0.18 | 1.8 | Hydrided for 50 h, $P_i$ = 32.7 Torr |
| ox1205-C1 | 26 | 4.5 | 1.85 | 19.5 | Hydrided for 50 h, $P_i$ = 32.8 Torr |
| ox1205-U2 | 319 | 8.9 | 0.14 | 1.4 | Hydrided for 50 h, $P_i$ = 51.1 Torr |
| ox1205-C2 | 24 | 4.6 | 1.88 | 19.7 | Hydrided for 50 h, $P_i$ = 51.0 Torr |
| ox1205-U3 | 755 | 8.6 | 0.04 | 0.4 | Hydrided for 51 h, $P_i$ = 71.3 Torr |
| ox1205-C3 | 32 | 4.5 | 1.86 | 19.6 | Hydrided for 51 h, $P_i$ = 71.3 Torr |

* Hydrogen contents were measured before oxidation tests. $P_i$ is initial pressure.

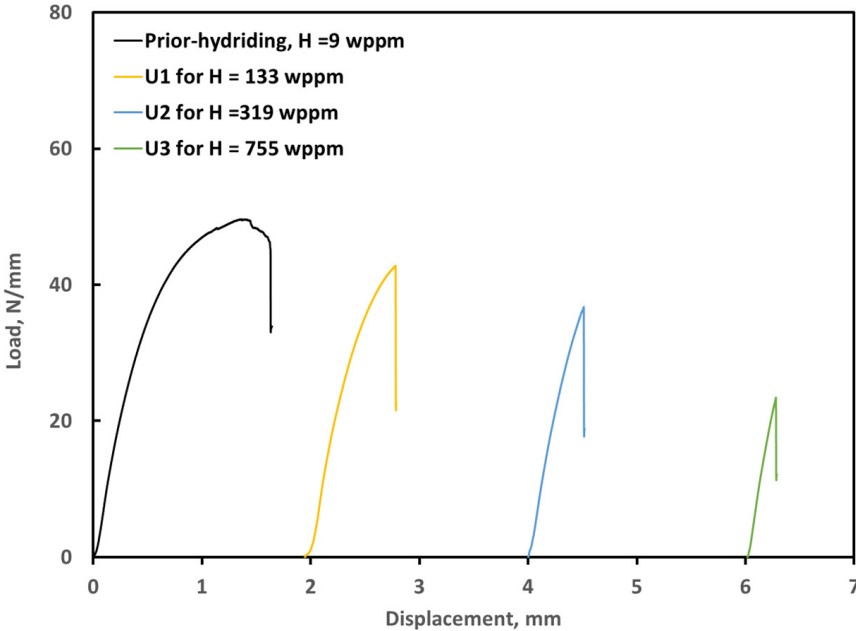

**Figure 12.** Ring-compression test load–displacement data of uncoated Zircaloy-4 after hydriding at 425 °C for 0–71 h and oxidation at 1205 °C for 100 s.

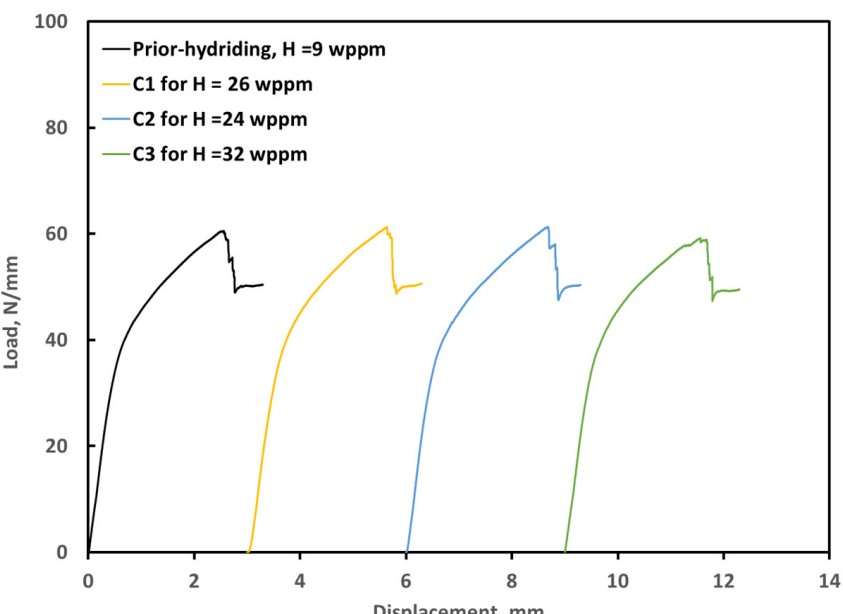

**Figure 13.** Ring-compression test load–displacement data of Cr-coated Zircaloy-4 after hydriding at 425 °C for 0–71 h and oxidation at 1205 °C for 100 s.

**Table 6.** RCT results for coated and uncoated Zircaloy-4 oxidized in steam at 1000 °C for 3000 s.

| Sample | Measured H * (wppm) | Weight Gain (mg/cm$^2$) | Offset Displacement (mm) | Offset Strain (%) | Comments |
|---|---|---|---|---|---|
| ox1000-U0 | 10 | 12.0 | 0.4 | 4.2 | As-polished |
| ox1000-C0 | 9 | 5.9 | 2.5 | 26.4 | As-coated |
| ox1000-U1 | 143 | 11.8 | 0.27 | 2.8 | Hydrided for 50 h, $P_i$ = 32.7 Torr |
| ox1000-C1 | 26 | 5.9 | 2.43 | 25.6 | Hydrided for 50 h, $P_i$ = 32.8 Torr |
| ox1000-U2 | 319 | 12.1 | 0.15 | 1.6 | Hydrided for 50 h, $P_i$ = 51.1 Torr |
| ox1000-C2 | 24 | 6.1 | 2.28 | 24.0 | Hydrided for 50 h, $P_i$ = 51.0 Torr |
| ox1000-U3 | 755 | 12.3 | 0.03 | 0.3 | Hydrided for 51 h, $P_i$ = 71.3 Torr |
| ox1000-C3 | 32 | 6.0 | 2.51 | 26.4 | Hydrided for 51 h, $P_i$ = 71.3 Torr |

* Hydrogen contents were measured before oxidation tests. $P_i$ is initial pressure.

Figure 14 shows sample ox1000-U2 and Figure 15 shows sample ox1000-C2 after oxidation and post ring compression testing. The micrograph in the center with a blue frame shows an overview of the sample. The surrounding micrographs are taken from that micrograph's position relative to the center overview image.

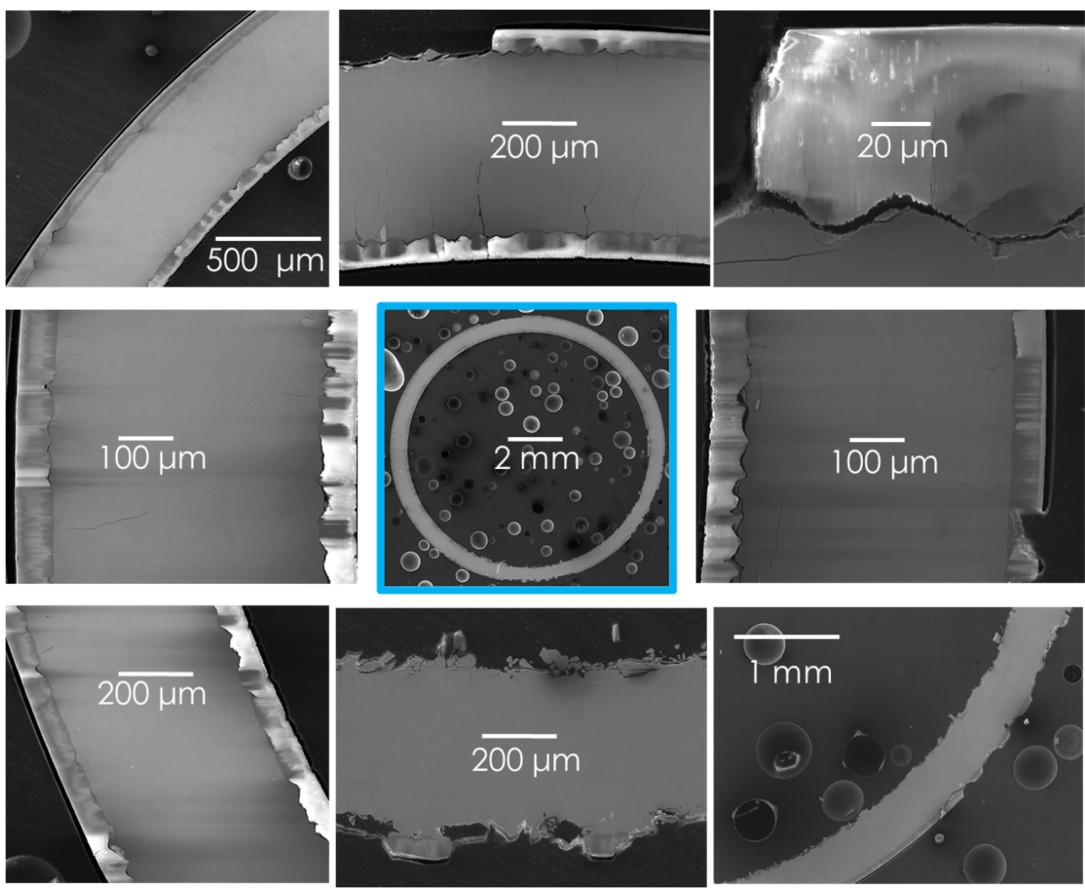

**Figure 14.** SEM images of sample ox1000-U2 after ring-compression test load displacement. Sample overview is in the middle (blue frame), and higher magnified areas are shown on the outside.

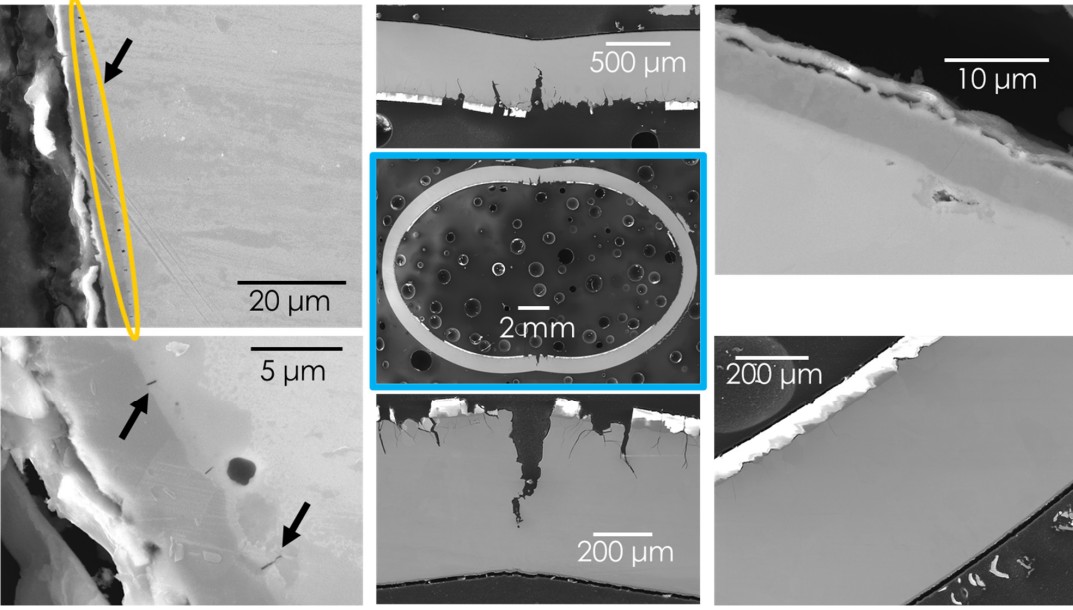

**Figure 15.** SEM images of sample ox1000-C2 after ring-compression test load displacement. Sample overview of the sample is in the middle (blue frame), and higher magnified areas are shown on the outside. The orange ellipse and black arrows point to crack initiation starting from the interface between coating and cladding in a radial direction at 2, 4, 8, and 10 o'clock positions of the shown sample in the center.

Figure 14 clearly shows that the inner and outer diameter have formed equally thick oxide layers of around 65 to 80 μm. In areas where the pressure was applied (12 and 6 o'clock positions) during the RCT, the outside oxide layer is crushed and delaminated. The oxide layer at the inner diameter in those positions was exposed to tensile stress, and several cracks through the oxide layer continuing into the Zircaloy-4 cladding with partial delamination of the oxide layer are observed. The reported ductility decrease can be explained by an oxide layer covering around 20% of the diameter of the entire sample. Crack initiation in the brittle oxide layer followed by a brittle failure causes the sample to exhibit poor ductility.

Figure 15 shows the coated sample after compression testing. More deformation is clearly visible in comparison with the uncoated sample. The oxide layer at the inner diameter exhibits a similar thickness to the uncoated condition. However, the outer diameter with around 7 μm coating thickness offers protection against corrosion. The Cr HiPMS coating has remained adherent to the cladding substrate. The surface of Cr has oxidized, and delamination between $Cr_2O_3$ and Cr is visible. The coating interface is mostly undamaged, but small radial cracks forming at the interlayer indicate that the cracks form under stress, which starts from the brittle laves phase [20]. The laves phase formed at elevated temperatures between the Cr coating and the Zircaloy-4 cladding [21]. This behavior is consistent with observations of high temperature oxidation of Cr-coated Zr cladding reported elsewhere [22,23]. Bending tests performed by other researchers have shown that cracks initiate from the brittle intermetallic in a similar way [21].

This work focused on unirradiated materials, and the selection of future candidate materials must consider other relevant properties that affect in-reactor materials performance. More studies on the irradiation performance of the coated cladding materials are under way.

## 4. Conclusions

Post quench ductility studies were conducted with the prehydrided coated and uncoated Zircaloy-4 samples after oxidation at temperatures of 1000 and 1205 °C. These studies determined ductility under one-sided hydriding followed by two-sided cladding steam exposure. The following conclusions were drawn from the results:

Coated and uncoated Zircaloy-4 tubing specimens were one-sided hydrided in a tube furnace filled with pure hydrogen gas at 425 °C up to 755 wppm of hydrogen. Hydrogen content of uncoated tubes increases with increasing test time and initial pressures. However, the coated specimens hydrided under the same experimental conditions exhibited very little change in hydrogen content. Coated specimens experienced slower oxidation under exposure to steam: oxygen pickup was 50% lower than that of the uncoated specimens tested under the same conditions. No zirconium oxide layer was observed on the coating surface, indicating that the coating provides significant protection. Ring compression testing for the uncoated Zircaloy-4 specimens showed that displacements and maximum loads gradually decrease with increasing hydrogen content. The coated specimens exhibited very little change in the load–displacement curves between the as-fabricated and hydrided specimens. Coated specimens remained very ductile after being oxidized at 1205 °C for 100 s and at 1000 °C for 3000 s, indicating that the coating provides significant protection from hydriding and oxidation at high temperatures.

**Author Contributions:** Y.Y.: Experimental data collection & analysis, original draft, editing; T.G.: Experimental data collection & analysis, writing, editing; A.T.N.: Conceptualization; writing, editing. All authors have read and agreed to the published version of the manuscript.

**Funding:** This research was funded by United States Department of Energy, grant number DE-AC05-00OR22725.

**Data Availability Statement:** The data presented in this study are available on request from the corresponding author. The data are not publicly available due to ongoing research in this area.

**Acknowledgments:** This work was sponsored by DOE-NE Advanced Fuel Campaign (AFC). We would like to express our appreciation to Caitlin Duggan for her help on metallographic mount preparation and examinations. We would like thank M. Ridley and B. Garrison for providing their comments in the manuscript.

**Conflicts of Interest:** The authors declare no conflict of interest.

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
