# Peer review of "Hydriding, Oxidation, and Ductility Evaluation of Cr-Coated Zircaloy-4 Tubing"

_metals, doi:10.3390/met12121998_

Round 1

Reviewer 1 Report

Dear authors,

the problem of accident-tolerant fuel assemblies is really important for nuclear power plants reliability, and many specialists are solving this task by some kinds of coatings. This paper seems to provide a good way to increase the resistance to BDB accidents.

But at the same time the paper needs some additional editing before publication.

Section 2 Materials and Methods:

1) The composition of Zr alloy from the standard could be listed in the paper but you shall also give the actual composition of tubes under study.

2) The first sentence of Results and Discussion section "The Zircalloy-4..." could be deleted, and next sentences (lines 142-156) should be moved to Section 2.

Section 4 Summary is called typically as Conclusions. On my mind it should be significantly shortened. It is not an abstract.
 I can suppose that you shall summarise only positive influence of Cr coating and of course with the last point about applicability of the proposed approach in real reactor conditions.

And, finally, please correct the font size on lines 51-54.

Author Response

1) The composition of Zr alloy from the standard could be listed in the paper but you shall also give the actual composition of tubes under study.

Yes. We changed Table 1 to give the actual composition of tubes under study.

2) The first sentence of Results and Discussion section "The Zircalloy-4..." could be deleted, and next sentences (lines 142-156) should be moved to Section 2.

Yes. It’s done.

Section 4 Summary is called typically as Conclusions. On my mind it should be significantly shortened. It is not an abstract.

Yes. It’s done.

And, finally, please correct the font size on lines 51-54.

Yes. It’s done.

Reviewer 2 Report

The authors have done an impressive job using a variety of experimental techniques. The methods are well described and provide meaningful results needed to address the question posed in the introduction. However, the description of the results could be improved, in particular the number of figures and tables could be shortened by combining some of them or omitting them because they are not necessary (such as Figure 8). Overall, this is a well-written paper that raises important questions and provides relevant results and insights into the hydrogenation and oxidation of coating materials.

Author Response

The authors have done an impressive job using a variety of experimental techniques. The methods are well described and provide meaningful results needed to address the question posed in the introduction. However, the description of the results could be improved, in particular the number of figures and tables could be shortened by combining some of them or omitting them because they are not necessary (such as Figure 8). Overall, this is a well-written paper that raises important questions and provides relevant results and insights into the hydrogenation and oxidation of coating materials.

Yes. We removed Figure 8.

Reviewer 3 Report

The article reviews the properties of chromium-coated Zircaloy-4, which have been tested to study the effects of hydrogenation, oxidation, and post-quench ductility behavior on coated Zr-shell. In general, this article corresponds to the subject of the declared journal and can be accepted for publication after the authors answer a number of questions that the reviewer had while reading it.

1. In general, this direction is quite interesting, but in the abstract, the authors should provide more details about the novelty and practical significance of the work.

2. The oxidation experiment requires clarification, how exactly such a sharp heating and subsequent rapid cooling was carried out, what was the basis for choosing the oxidation time and annealing temperature?

3. The authors should provide more data on the change in the degree of surface roughness as a result of polishing the samples, as well as the need for this procedure.

4. The authors should, as an example, give a comparative analysis of the studied samples with irradiated samples.

5. In addition, the authors should consider submitting mapping or EBSD results to better represent oxidation and hydrogenation processes.

6. The data shown in Figure 7 require a more detailed analysis, the authors should pay more attention to a detailed representation of the resulting microcracks and pores, as well as explain the mechanism of their occurrence.

7. There are a number of typos in the text of the article, which the authors should get rid of before accepting the article for publication.

Author Response

  1. In general, this direction is quite interesting, but in the abstract, the authors should provide more details about the novelty and practical significance of the work.

Yes. We added a few sentences in the abstract to address it.

  1. The oxidation experiment requires clarification, how exactly such a sharp heating and subsequent rapid cooling was carried out, what was the basis for choosing the oxidation time and annealing temperature?

The test parameters, such as the heating rate, cooling rate, hold temperature, and water quench temperature were carefully selected based on a guideline provided by the US NRC. The test time was determined by oxygen pickup near the ductile-to-brittle transition point of the oxidized specimens. We provided the NRC’s guideline as a reference [15].

  1. The authors should provide more data on the change in the degree of surface roughness as a result of polishing the samples, as well as the need for this procedure.

In Section 2 (line 65-77), we have provided a paragraph to address the roughness issue “The surface roughness of the tube was measured with a Mahr MarTalk profilometer using the MarWin software package. The tubes were measured in different areas to identify possible differences. Roughness measurements were performed measured 11200 points across a 4 mm length profile on the tube surface. Even though the average surface roughness was within the specified limits of 0.81 μm, maximum roughness peaks of approximately 2 to 4 μm or more were observed near scratches on the surface. These scratches were concerning with respect to their potential to affect coating adherence or cause cracks or discontinuities in the coating. Therefore, the as-received tubes were subsequently from a 30 μm to a 3 μm polishing paper before coating, hydriding, and steam oxidations. Polishing improved the surface roughness from an average roughness of around 0.36 to around 0.1 μm and decreased maximum roughness from 4 to around 1 μm. Surfaces of as-received and polished Zircaloy-4 tubes are shown in Figure 1.”

  1. The authors should, as an example, give a comparative analysis of the studied samples with irradiated samples.

Yes. It’s very important to study the irradiated samples. We appreciate the reviewer’s comment. We don’t have the irradiated specimens at the moment, but plan to irradiate our own material in FY23. The result will be presented in the future. Note: it will take 1-2 years to obtain irradiated samples.

  1. In addition, the authors should consider submitting mapping or EBSD results to better represent oxidation and hydrogenation processes.

The hydrogen content in this work is quite low: <750 wppm. The hydride in these samples is not enough to be visible in the EBSD. Based on our experience, the optical images in Figure 7 provide the best contrast of hydrides in hydrided Zr cladding. In addition, hydrogen/hydride is not visible by the SEM EDS or mapping.  We actually performed mapping (see attached), and believe SEM images in secondary electron mode (see Figure 2) provide much better resolution. 

In addition, this work is to address that the Cr-coating can provide protection against the hydriding and oxidation, which have been confirmed by ring compression testing. We appreciate reviewers’ comments for in-depth microstructural evaluations on post hydriding and oxidation Cr coated samples. We plan to do additional study of TEM and SEM to further investigate the interaction between the coating layer and Zr matrix, the result of which will be reported in a separated paper.

  1. The data shown in Figure 7 require a more detailed analysis, the authors should pay more attention to a detailed representation of the resulting microcracks and pores, as well as explain the mechanism of their occurrence.

The dark lines in Figure 7 are not microcracks or pores. They are hydrides after hydriding experiment. We added a reference about the hydrided formed in Zry-4. We also added a few sentences to the description of Figure 7 to avoid any confusion in the manuscript.

  1. There are a number of typos in the text of the article, which the authors should get rid of before accepting the article for publication.

Thanks. Our technical editor reviewed the manuscript.

Round 2

Reviewer 3 Report

The authors have completely corrected all the comments indicated by the reviewer, the article can be accepted for publication.